# Maternal morbidity profile in hospitalizations in the Unified Health System in São Paulo, Brazil: Analysis using data mining, 2014 to 2019

Rejane Sobrino Pinheiro[1]*, Marcos Augusto Bastos Dias[2],
Luís Guilherme Buteri Alves[1], Luís Carlos Torres Guillén[3], Lana dos Santos Meijinhos[4],
Natália Santana Paiva[1], Valeria Saraceni[4], Claudia Medina Coeli[1],
Rosa Maria Soares Madeira Domingues[5]

1 Universidade Federal do Rio de Janeiro, Instituto de Estudos em Saúde Coletiva, Rio de Janeiro, Brazil, 2 Fundação Oswaldo Cruz, Instituto Nacional de Saúde da Mulher, da Criança e do Adolescente Fernandes Figueira, Rio de Janeiro, Brazil, 3 Secretaria Municipal de Saúde de Niteroi, Rio de Janeiro, Brazil, 4 Secretaria Municipal de Saúde do Rio de Janeiro, Rio de Janeiro, Brazil, 5 Fundação Oswaldo Cruz, Instituto Nacional de Infectologia Evandro Chagas, Rio de Janeiro, Brazil

* rejane@iesc.ufrj.br

## Abstract

In Brazil, approximately 80% of births are publicly funded and are registered in the Hospital Information System of the Unified Health System (SIH/SUS). The SIH/SUS has morbidity data, but its use for maternal morbidity surveillance is still limited. The objective of this study was to apply data mining techniques to identify diagnoses registered in the SIH/SUS that alone or together are associated with maternal death. Obstetric hospitalizations at SIH/SUS of women aged 10–49 years from the largest state in Brazil (São Paulo) from 2014 to 2019 were analysed. Diagnoses were classified into 12 groups, adapting the WHO's classification proposal of maternal deaths. Groups of diagnoses that were related to maternal death were identified using association rules with the Apriori algorithm. Of the 2,742,467 hospitalizations, 831 (0.03%) resulted in death. The most frequent diagnoses associated with death, alone or in combination, were non-obstetric complications (62.0%). Pregnancy-specific hypertensive complications (20.1%), pregnancy-related infections (19.9%) and haemorrhages (13.6%) were also present in hospitalizations resulting in death. The risk of death was more significant for non-obstetric complications, unknown causes and external causes, with lifts above 10. The risk of death was at least three times greater for groups of diagnoses related to the most frequent causes of maternal death in the country (complications of hypertension, infections, and haemorrhages). Uncomplicated abortion, gestational diabetes and other causes of non-obstetric hospitalization were only associated with maternal death when together with other diagnoses. The study reinforces the data mining technique as an innovative approach in the study of maternal mortality in the country. The results also reaffirm the importance of hypertensive, haemorrhagic and infectious causes for maternal death and highlight the

**Data availability statement:** The file analysed in the manuscript can be accessed in https://doi.org/10.7303/syn69944143.

**Funding:** This work was supported by the Bill & Melinda Gates Foundation [INV-027961] and the Brazilian Ministry of Health/DECIT/CNPq [445116/2020-0]. RMSMD received the grants. Additionally, RSP [316755/2021-4] and CMC [306668/2023-8] were partially supported by research fellowship grants from the Conselho Nacional de Desenvolvimento Científico e Tecnológico (CNPq). The funders had no role in study design, data collection and analysis, decision to publish, or preparation of the manuscript.

**Competing interests:** The authors have declared that no competing interests exist.

relevance of non-obstetric causes for the surveillance of maternal morbidity since women with these diagnoses, alone or associated with other complications, had a higher risk of death.

## Introduction

Brazil has a high maternal mortality ratio (MMR), which is greater than 50 per 100,000 live births [1]. In the period 2009–2020, the leading causes of maternal death were direct obstetric causes, predominantly hypertension, haemorrhage, puerperal infection, and abortion [1]. Among the non-obstetric causes, the main ones are diseases of the circulatory system, respiratory system diseases, and maternal infectious and parasitic diseases [1].

Reducing maternal mortality depends on detecting diseases and complications developed during pregnancy, childbirth, and the postpartum period and the consequent response of the health system to manage them, covering aspects of patient safety and the ability to reduce negative consequences. Surveillance of maternal morbidity associated with the worsening of women's conditions during pregnancy, childbirth, or the postpartum period allows for a deeper understanding of preventable maternal deaths. It contributes to detecting deficiencies in health systems to reorient and prioritize actions.

The investigation of maternal deaths was established in Brazil in 2009 and is essential for a better understanding of the causes and determinants of maternal deaths [2]. However, although the MMR is high in the country, maternal deaths are infrequent events, especially in the context of health services, and it is recommended that maternal morbidity be studied in a complementary manner to develop strategies to improve obstetric care and reduce maternal mortality [3].

In Brazil, approximately 80% of births are publicly funded [4] and are registered in the Hospital Information System of the Unified Health System (from the Portuguese name Sistema de Informação Hospitalar do Sistema Único de Saúde – SIH/SUS). Each hospital admission record has several fields that allow the reporting of different obstetric diagnoses and related comorbidities.

Studies on maternal health during pregnancy, delivery, and postpartum have traditionally examined the isolated effects of various causes on maternal mortality. However, morbidities may operate independently or exhibit interconnections that impact women's health differently. Interpreting statistical models becomes even more challenging when analyzing rare events when numerous morbidities might interact or occur in isolation, leading to a high number of independent variables. By employing data mining techniques, analysis strategies can yield novel insights from extensive variable sets, especially those contributing to an increased risk of maternal death. Understanding how distinct morbidity profiles relate to higher risks of maternal death is crucial for supporting prevention efforts and identifying early warnings of adverse prognosis. Thus, the objective of this study was to apply data mining techniques to identify diagnoses registered in the SIH/SUS that alone or together are associated with maternal death.

## Method

### Data source

We analysed the obstetric hospitalizations of women aged 10–49 years admitted to São Paulo State's public or private hospitals funded by the Unified Health System (SUS) and registered in the SIH/SUS between January 2014 and December 2019. São Paulo is the largest state in the country and has a wide, publicly funded network of health services trained in childbirth and postpartum care. The definition of the study period was due to the COVID-19 pandemic, which began in March 2020. The COVID-19 pandemic changed the maternal mortality profile in the country, with an increase in the maternal mortality ratio and in the proportion of deaths from non-obstetric causes, therefore being an atypical period [5,6]. We opted, therefore, to gather data from the more recent years before the COVID-19 pandemic, characterized by a largely stable maternal death pattern, to construct a dataset with an adequate number of deaths for robust data analysist.

In the SIH/SUS, a hospital admission may consist of one or more claim records, called Hospital Admission Authorization (AIH- from Portuguese, *Autorização de Internação Hospitalar*). When AIH discharge was a hospital stay or an interhospital transfer, the subsequent record (within a one-day maximum difference between the admission date and the previous discharge date) was searched via a deterministic record linkage algorithm using variables available in deidentified databases: the hospital code, the municipality of residence, and the patient's date of birth. All AIH records of the same patient identified by the algorithm as belonging to the same hospitalization were grouped and analysed as one hospital episode of care. More information about the development of this algorithm is available at Coeli et al. [7]. We selected hospital episodes where at least one AIH presented: i) an obstetric diagnosis (chapter XV of the International Classification of Disease – ICD X) in the principal or any of the 11 secondary diagnostic fields; or ii) an obstetric principal care procedure [8].

Among these episodes of obstetric hospitalization, we selected those whose reason for discharge was recorded as "discharged alive" or "in-hospital death" as the objective was to analyse the profile of diagnoses registered in the SIH/SUS associated with maternal death. We excluded 222 hospital episodes with the discharge coded recorded as "in-hospital death" because we could not identify these deaths in the Mortality Information System (SIM) database via a deterministic matching algorithm.

The final database analysed included 2,742,467 hospital records. Most of them (72.6%) had only one primary diagnosis and only one AIH (98.9%). The SIH/SUS and SIM anonymized databases are publicly available at https://datasus.saude.gov.br/transferencia-de-arquivos/.

### Morbidity classification

We classified each diagnosis into 12 groups, some of which were divided into subgroups (S1 Table). Our classification was based on the one proposed by the International Classification of Diseases, 10th revision—maternal ICD of the World Health Organization (WHO) [9], which classifies maternal deaths. As the AIH database contains data on morbidity, not causes of mortality, the WHO maternal ICD groups were adapted to capture all diagnoses recorded during hospitalization, not just causes of death. In this adaptation, proposed by an obstetrician of our research team, ICD codes that are not considered causes of death, but are causes of morbidity and hospital admissions, were added to the existing maternal ICD groups, while new groups that are exclusively related to maternal morbidity were created. The proposed adaptation is displayed in S1 Table.

### Data mining

We used a data mining technique based on association rules to identify diagnosis groups associated with hospital discharge (alive or death). The associations were assessed by considering both isolated and combined diagnoses.

This technique has been used in shopping basket analysis to identify which items in the basket are purchased together [10,11]. Association rules are based on combinatorial analysis of sets of different items belonging to a database. An item

represents a category of a variable. For example, male is one item, and female is another item. The rule is based on the relationship of a set of items called antecedents (represented by X) to another set of items called consequents (represented by Y). The relationship can be expressed by a conditional sentence of the type IF (X) THEN (Y), symbolized by X→Y. An association rule depicts the relationship between items, checking whether items $x_1$, $x_2$,...,$x_n$ are associated with items $y_1$, $y_2$,...$y_k$ [10].

The rules are generated considering all possible combinations of items, and metrics related to the association's frequency and "strength" are calculated. The most commonly used metrics are support, confidence, and lift, which are used to select the most relevant rules [12].

Support is related to the frequency of a database profile (set of items). The support of a rule, Supp(X→Y), is the proportion in which the antecedents and consequents are found concomitantly in the database (Eq. 1). The confidence of a rule measures the proportion in which the consequent appears in the set of records that have the profile given by the antecedents (Eq. 2). Lift measures the dependence between the items of a rule, pointing to the strength of the association observed between the consequent and the antecedent, compared with the situation in which the consequent and the antecedent would represent independent events (Eq. 3) [10–12].

In the analysed database, death in hospitals was a very infrequent event, so the minimum support and confidence were set at $1.0 \times 10^{-10}$. This allowed the generation of infrequent rules to be mined later via the metrics described by Eq. (1) to (3).

$$\text{Supp}(X \rightarrow Y) = \frac{\text{Supp}(X \cap Y)}{n} = \frac{n^o \text{ records with the profile } (X \cap Y)}{n}, \tag{1}$$

where *n* is the number of records in the database.

$$\text{Conf}(X \rightarrow Y) = \frac{\text{Supp}(X \cap Y)}{\text{Supp}(X)} \tag{2}$$

$$\text{Lift}(X \rightarrow Y) = \frac{\text{Supp}(X \cap Y)}{\text{Supp}(X).\text{Supp}(Y)} = \frac{\text{Conf}(X \rightarrow Y)}{\text{Supp}(Y)} \tag{3}$$

We used the Apriori algorithm [11] implemented in the arules R package (version 1.6–8) (https://www.rdocumentation.org/packages/arules/versions/1.6-8). Rules with a lift lower than 1.1 and redundant rules were eliminated via the is.redundant function and the confidence metric (https://rdrr.io/cran/arules). A redundant rule has additional items but with confidence equal to or lower than that obtained by the original rule. The rules that contained the most relevant diagnosis groups were initially selected via support, confidence, and lift metrics. Because in obstetric hospitalization, in-hospital death is a rare event, we analysed infrequent rules when they occurred at least four times in the database. Finally, we excluded rules with diagnoses of "group 10 (childbirth)", as they are not relevant to the morbidity analysis.

We analysed databases that were deidentified and made publicly available (open access). According to the Brazilian National Health Council ethics resolution n° 510/2016 (April 7, 2016), research ethics committee approval was waived.

## Results

Among the 2,742,467 hospital episodes evaluated, 831 (0.03%) had a discharge code recorded as an in-hospital death.

The algorithm generated 744 rules. We eliminated all rules with "discharge alive" as a consequent because they had a lift less than 1.1. Our analysis then focused on 120 (14.8%) rules in which "in-hospital death" was the consequent and the lift was above 1.1. Among these 120 rules, 16 presented only one diagnosis group/subgroup in the antecedent (Table 1).

**Table 1. Frequency of diagnostic groups in obstetric hospitalizations of women aged 10-49 years with in-hospital death in the SIH/SUS. State of São Paulo, Brazil, 2014–2019.**

| Group | Subgroup | Support | Confidence | Lift | Number of cases | Group total |
|---|---|---|---|---|---|---|
| G1 – Abortion | 1b – Abortion with complications | 1,39E-05 | 0.001 | 3.029 | 38 | 38 |
| G2 – Hypertensive disorders | 2a - Pregnancy-specific hypertension | 6,09E-05 | 0.001 | 4.376 | 167 | 201 |
| | 2b - Chronic hypertension | 1,24E-05 | 0.001 | 4.2177 | 34 | |
| G3 - Obstetric haemorrhages | | 4,12E-05 | 0.001 | 4.245 | 113 | 113 |
| G4 - Infections | 4a – Pregnancy-related infections | 6,02E-05 | 0.002 | 7.510 | 165 | 187 |
| | 4b – Infections not related to pregnancy | 8,02E-06 | 0.002 | 6.300 | 22 | |
| G5 - Other obstetric complications | | 2,88E-05 | 0.002 | 5.774 | 79 | 79 |
| Group 7 – Diabetes | 7b - Non-gestational diabetes | 4,74E-06 | 0.0004 | 1.310 | 13 | 13 |
| G8 - Non-obstetric complications | 8a - Respiratory system diseases complicating pregnancy, childbirth and the postpartum period | 4,81E-05 | 0.022 | 73.018 | 132 | 515 |
| | 8b - Diseases of the digestive system complicating pregnancy, childbirth and the postpartum period | 1,79E-05 | 0.015 | 51.045 | 49 | |
| | 8c – Diseases of the circulatory system complicating pregnancy, childbirth and the postpartum period | 4,41E-05 | 0.014 | 47.090 | 121 | |
| | 8d - Anemia complicating pregnancy, childbirth and the postpartum period | 5,83E-06 | 0.003 | 9.183 | 16 | |
| | 8e - Other non-obstetric causes complicating pregnancy, childbirth and the postpartum period | 7,18E-05 | 0.003 | 10.053 | 197 | |
| G9 – Unknown causes | | 4,74E-06 | 0.236 | 780.048 | 13 | 13 |
| G12 – External causes | | 8,02E-06 | 0.006 | 20.190 | 22 | 22 |
| G13 - Group others* | | 1,20E-05 | 0.001 | 2.647 | 33 | 33 |

**\*Note:** Group "Others" includes all ICD codes that do not fall into the previously specified categories.

The most frequent diagnosis group was non-obstetric complications (group 8), which were present in 62.0% (515) of hospital episodes that ended in death. In this group, "Other non-obstetric diseases"(group 8e) appeared in 23.7% of hospital episodes (197), "Diseases of the respiratory system " (group 8a) in 15.9% (132), and "Diseases of the circulatory system" (group 8c) in 14.5% (121). Pregnancy-specific hypertension (group 2a) was present in 20.1% (167) of hospitalizations resulting in death, pregnancy-related infections (group 4a) in 19.9% (165), obstetric haemorrhages (group 3) in 13.6% (113), and other obstetric complications (group 5) in 9.5% (79). The other remaining groups were present in 14.3% of hospital episodes that ended in death: group 1b (abortion with complications), group 2b (chronic hypertension), group 13 "others" (other causes not classified in the previous groups), group 4b infections not related to pregnancy) and group 12 (external causes) (Table 1).

The diagnosis groups with the highest risk of death, with lifts above 10, were non-obstetric complications (group 8), unknown causes (group 9), and external causes (group 12) (Table 1). Hypertensive disorders (group 2), infections (group 4), and obstetric haemorrhages (group 3), which correspond to the most frequent causes of maternal death in mortality statistics, presented lifts above 3 (Table 1).

Sixty-three rules combined two groups/subgroups, whereas 40 combined three (Table 2). Only one rule presented four diagnosis groups/subgroups in the antecedent. Diagnostic groups 1a (Abortion without complications), 7a (gestational diabetes), and 11 (other obstetric conditions) alone were not associated with maternal death. However, they appeared to be associated with in-hospital death when accompanied by other diagnoses. Abortion without complications (group 1a)

**Table 2. Most frequent combinations of diagnostic groups in hospitalizations that result in death. State of São Paulo, Brazil, 2014–2019.**

| 1º Diagnostic group/subgroup | 2º Diagnostic group/subgroup | 3º Diagnostic group/subgroup | Confidence | Lift | Number of cases |
|---|---|---|---|---|---|
| 1a – Abortion without complications | 4a - Pregnancy-related infections | | 0.044 | 143.487 | 17 |
| | | 8a - Diseases of the respiratory system complicating pregnancy, childbirth and the postpartum period | 0.750 | 2475.151 | 6 |
| | | 8e - Other non-obstetric diseases complicating pregnancy, childbirth and the postpartum period | 0.214 | 707.186 | 6 |
| | 8a - Diseases of the respiratory system complicating pregnancy, childbirth and the postpartum period | | 0.109 | 359.428 | 11 |
| | | 8e - Other non-obstetric diseases complicating pregnancy, childbirth and the postpartum period | 0.357 | 1178.643 | 5 |
| | 8c - Diseases of the circulatory system complicating pregnancy, childbirth and the postpartum period | 8e - Other non-obstetric diseases complicating pregnancy, childbirth and the postpartum period | 0.333 | 1100.067 | 4 |
| | 8e - Other non-obstetric diseases complicating pregnancy, childbirth and the postpartum period | | 0.005 | 15.825 | 11 |
| 1b – Abortion with complications | 4a - Pregnancy-related infections | | 0.045 | 149.523 | 14 |
| | | 8a - Diseases of the respiratory system complicating pregnancy, childbirth and the postpartum period | 0.500 | 1650.101 | 5 |
| | | 8e - Other non-obstetric diseases complicating pregnancy, childbirth and the postpartum period | 0.231 | 761.585 | 6 |
| | 8a - Diseases of the respiratory system complicating pregnancy, childbirth and the postpartum period | | 0.117 | 385.023 | 7 |
| | 8c - Diseases of the circulatory system complicating pregnancy, childbirth and the postpartum period | | 0.096 | 317.327 | 5 |
| | 8e - Other non-obstetric diseases complicating pregnancy, childbirth and the postpartum period | | 0.006 | 21.264 | 10 |
| 2a - Pregnancy-specific hypertension | 2b - Chronic hypertension | | 0.003 | 9.054 | 8 |
| | | 8c - Diseases of the circulatory system complicating pregnancy, childbirth and the postpartum period | 0.174 | 573.948 | 4 |
| | 3 – Obstetric haemorrhages | | 0.026 | 85.943 | 15 |
| | | 5 – Other obstetric complications | 0.182 | 600.037 | 5 |
| | | 11 – Other obstetric conditions | 0.036 | 118.926 | 4 |
| | 4a - Pregnancy-related infections | | 0.011 | 34.521 | 20 |
| | | 8a - Diseases of the respiratory system complicating pregnancy, childbirth and the postpartum period | 0.159 | 525.032 | 7 |
| | | 8b – Diseases of the digestive system complicating pregnancy, childbirth and the postpartum period | 0.238 | 785.762 | 5 |
| | | 8c - Diseases of the circulatory system complicating pregnancy, childbirth and the postpartum period | 0.308 | 1015.447 | 4 |
| | | 8e - Other non-obstetric diseases complicating pregnancy, childbirth and the postpartum period | 0.031 | 101.858 | 5 |
| | | 11 – Other obstetric conditions | 0.011 | 34.593 | 5 |

*(Continued)*

**Table 2.** (Continued)

| 1º Diagnostic group/subgroup | 2º Diagnostic group/subgroup | 3º Diagnostic group/subgroup | Confidence | Lift | Number of cases |
|---|---|---|---|---|---|
| | 5 – Other obstetric complications | | 0.009 | 29.821 | 12 |
| | | 8c - Diseases of the circulatory system complicating pregnancy, childbirth and the postpartum period | 0.333 | 1100.067 | 4 |
| | | 8e - Other non-obstetric diseases complicating pregnancy, childbirth and the postpartum period | 0.100 | 330.020 | 6 |
| | 8a - Diseases of the respiratory system complicating pregnancy, childbirth and the postpartum period | | 0.055 | 181.330 | 25 |
| | | 8c - Diseases of the circulatory system complicating pregnancy, childbirth and the postpartum period | 0.292 | 962.559 | 7 |
| | | 8e - Other non-obstetric diseases complicating pregnancy, childbirth and the postpartum period | 0.152 | 500.030 | 10 |
| | 8b – Diseases of the digestive system complicating pregnancy, childbirth and the postpartum period | | 0.057 | 186.804 | 9 |
| | | 8e - Other non-obstetric diseases complicating pregnancy, childbirth and the postpartum period | 0.200 | 660.040 | 4 |
| | 8c - Diseases of the circulatory system complicating pregnancy, childbirth and the postpartum period | | 0.110 | 362.938 | 43 |
| | | 8e - Other non-obstetric diseases complicating pregnancy, childbirth and the postpartum period | 0.288 | 950.905 | 17 |
| | 8d – Anemia complicating pregnancy, childbirth and the postpartum period | | 0.011 | 37.185 | 4 |
| | 8e - Other non-obstetric diseases complicating pregnancy, childbirth and the postpartum period | | 0.011 | 36.640 | 42 |
| | | 11 – Other obstetric conditions | 0.013 | 42.356 | 12 |
| 2b - Chronic hypertension | 8a - Diseases of the respiratory system complicating pregnancy, childbirth and the postpartum period | | 0.030 | 99.404 | 5 |
| | 8c - Diseases of the circulatory system complicating pregnancy, childbirth and the postpartum period | | 0.032 | 105.186 | 8 |
| | 8e - Other non-obstetric diseases complicating pregnancy, childbirth and the postpartum period | | 0.006 | 18.799 | 9 |
| 3 – Obstetric Haemorrhages | 4a - Pregnancy-related infections | | 0.013 | 43.063 | 11 |
| | 5 – Other obstetric complications | | 0.019 | 62.583 | 15 |
| | | 8e - Other non-obstetric diseases complicating pregnancy, childbirth and the postpartum period | 0.194 | 638.749 | 6 |
| | | 11- Other obstetric conditions | 0.076 | 250.648 | 6 |
| | 8a - Diseases of the respiratory system complicating pregnancy, childbirth and the postpartum period | | 0.067 | 220.013 | 4 |
| | 8b – Diseases of the digestive system complicating pregnancy, childbirth and the postpartum period | | 0.068 | 223.742 | 4 |
| | 8c - Diseases of the circulatory system complicating pregnancy, childbirth and the postpartum period | | 0.037 | 122.230 | 5 |
| | 8e - Other non-obstetric diseases complicating pregnancy, childbirth and the postpartum period | | 0.013 | 42.014 | 20 |
| | | 11- Other obstetric conditions | 0.046 | 152.989 | 7 |
| | | 13 – Others | 0.500 | 1650.101 | 4 |
| | 11 – Other obstetric conditions | | 0.009 | 31.017 | 35 |
| | 13 – Others | | 0.085 | 281.724 | 7 |

*(Continued)*

**Table 2.** (Continued)

| 1º Diagnostic group/subgroup | 2º Diagnostic group/subgroup | 3º Diagnostic group/subgroup | Confidence | Lift | Number of cases |
|---|---|---|---|---|---|
| 4a- Pregnancy-related infections | 4b - Infections not related to pregnancy | | 0.021 | 70.051 | 9 |
| | | 8a - Diseases of the respiratory system complicating pregnancy, childbirth and the postpartum period | 0.333 | 1100.067 | 5 |
| | 5 – Other obstetric complications | | 0.018 | 60.140 | 16 |
| | | 8a - Diseases of the respiratory system complicating pregnancy, childbirth and the postpartum period | 0.211 | 694.779 | 4 |
| | | 8e - Other non-obstetric diseases complicating pregnancy, childbirth and the postpartum period | 0.121 | 398.300 | 7 |
| | 7b - Non-gestational diabetes | | 0.008 | 24.665 | 5 |
| | 8a - Diseases of the respiratory system complicating pregnancy, childbirth and the postpartum period | | 0.110 | 361.958 | 51 |
| | 8a - Diseases of the respiratory system complicating pregnancy, childbirth and the postpartum period | 8c - Diseases of the circulatory system complicating pregnancy, childbirth and the postpartum period | 0.250 | 825.050 | 6 |
| | | 8e - Other non-obstetric diseases complicating pregnancy, childbirth and the postpartum period | 0.278 | 916.723 | 20 |
| | 8b – Diseases of the digestive system complicating pregnancy, childbirth and the postpartum period | | 0.077 | 253.862 | 20 |
| | | 8e - Other non-obstetric diseases complicating pregnancy, childbirth and the postpartum period | 0.114 | 375.023 | 5 |
| | | 11 – Other obstetric conditions | 0.171 | 563.449 | 7 |
| | 8c - Diseases of the circulatory system complicating pregnancy, childbirth and the postpartum period | | 0.063 | 207.342 | 12 |
| | | 8e - Other non-obstetric diseases complicating pregnancy, childbirth and the postpartum period | 0.286 | 942.915 | 8 |
| | 8e - Other non-obstetric diseases complicating pregnancy, childbirth and the postpartum period | | 0.015 | 49.022 | 56 |
| | | 11 – Other obstetric conditions | 0.019 | 62.268 | 8 |
| | 11- Other obstetric conditions | | 0.003 | 10.446 | 27 |
| 4b - Other Infections Not Related to Pregnancy | 8a - Diseases of the respiratory system complicating pregnancy, childbirth and the postpartum period | | 0.066 | 217.118 | 10 |
| | 8e - Other non-obstetric diseases complicating pregnancy, childbirth and the postpartum period | | 0.011 | 37.624 | 7 |
| 5 – Other obstetric complications | 8a - Diseases of the respiratory system complicating pregnancy, childbirth and the postpartum period | | 0.077 | 253.862 | 12 |
| | | 8e - Other non-obstetric diseases complicating pregnancy, childbirth and the postpartum period | 0.222 | 733.378 | 4 |
| | 8b – Diseases of the digestive system complicating pregnancy, childbirth and the postpartum period | | 0.030 | 99.404 | 5 |
| | 8c - Diseases of the circulatory system complicating pregnancy, childbirth and the postpartum period | | 0.060 | 197.547 | 17 |
| | | 8e - Other non-obstetric diseases complicating pregnancy, childbirth and the postpartum period | 0.231 | 761.585 | 6 |
| | 8e - Other non-obstetric diseases complicating pregnancy, childbirth and the postpartum period | | 0.023 | 77.262 | 28 |
| | | 11 – Other obstetric conditions | 0.040 | 132.008 | 5 |
| | 11 – Other obstetric conditions | | 0.002 | 5.831 | 15 |
| | 12 – External causes | | 0.019 | 62.563 | 4 |

*(Continued)*

**Table 2.** (Continued)

| 1º Diagnostic group/subgroup | 2º Diagnostic group/subgroup | 3º Diagnostic group/subgroup | Confidence | Lift | Number of cases |
|---|---|---|---|---|---|
| 7a - Gestational diabetes | 11 – Other obstetric conditions | | 0.001 | 2.064 | 4 |
| 7b - Non-gestational diabetes | 8a - Diseases of the respiratory system complicating pregnancy, childbirth and the postpartum period | | 0.036 | 117.864 | 4 |
| 8a - Diseases of the respiratory system complicating pregnancy, childbirth and the postpartum period | 8b – Diseases of the digestive system complicating pregnancy, childbirth and the postpartum period | | 0.091 | 300.018 | 4 |
| | 8c- Diseases of the circulatory system complicating pregnancy, childbirth and the postpartum period | | 0.153 | 506.166 | 25 |
| | | 8e - Other non-obstetric diseases complicating pregnancy, childbirth and the postpartum period | 0.419 | 1383.955 | 13 |
| | | 11 – Other obstetric conditions | 0.182 | 600.037 | 4 |
| | 8d - Anemia complicating pregnancy, childbirth and the postpartum period | | 0.045 | 148.324 | 4 |
| | 8e - Other non-obstetric diseases complicating pregnancy, childbirth and the postpartum period | | 0.084 | 275.955 | 49 |
| 8b – Diseases of the digestive system complicating pregnancy, childbirth and the postpartum period | 8e - Other non-obstetric diseases complicating pregnancy, childbirth and the postpartum period | | 0.041 | 133.792 | 18 |
| | 11 – Other obstetric conditions | | 0.026 | 84.332 | 15 |
| | 12 – External causes | | 0.100 | 330.020 | 4 |
| 8c – Diseases of the circulatory system complicating pregnancy, childbirth and the postpartum period | 8e - Other non-obstetric diseases complicating pregnancy, childbirth and the postpartum period | | 0.06586 | 217.344 | 38 |
| | 11 – Other obstetric conditions | | 0.017 | 54.800 | 18 |
| | 12 – External causes | | 0.031 | 102.332 | 4 |
| 8d - Anemia complicating pregnancy, childbirth and the postpartum period | 8e - Other non-obstetric diseases complicating pregnancy, childbirth and the postpartum period | | 0.012 | 37.933 | 5 |
| | 11 – Other obstetric conditions | | 0.005 | 16.731 | 8 |
| 8e - Other non-obstetric diseases complicating pregnancy, childbirth and the postpartum period | 12 – External causes | | 0.019 | 62.2684 | 7 |
| | 13 – Others | | 0.053 | 175.011 | 7 |
| 9 – Unknown causes | 11 – Other obstetric conditions | | 0.455 | 1500.091 | 5 |
| 11 – Other obstetric conditions | 12 – External causes | | 0.010 | 33.002 | 7 |
| | 13 – Others | | 0.008 | 26.722 | 4 |

appeared in rules together with pregnancy related infections (group 4a), diseases of the respiratory system (group 8a) or diseases of the circulatory system (group 8c), or other non-obstetric diseases (group 8e) (Table 2). Other causes of non-obstetric hospitalization (group 11) appeared together with gestational diabetes (group 7a) or other obstetric complications (group 5), external causes (group 12), and other diagnoses (group 13) in hospitalizations that ended in death (Table 2).

Furthermore, [Table 2](link) (and [S1](link) and [S2 Figs](link)) show that when the diagnosis groups were combined with other diagnosis groups/subgroups, groups related to complications, especially the subgroup 8 (non-obstetric complications), appeared more frequently. Only groups 9 (unknown causes) and 11 (other obstetric conditions) did not occur with group 8. In general, when a subgroup was associated with group 11, the lifts were the smallest among all other combinations of that subgroup.

When a rule presents three diagnosis groups/subgroups in the antecedent, the third diagnosis group frequently includes complications from other non-obstetric causes (group 8e), increasing the risk of death by 5–10 times, as observed by the lift values. These rules included women with abortion with and without complications, Pregnancy-specific hypertension, obstetric haemorrhages, pregnancy-related infections,other obstetric complications and – Diseases of the respiratory system complicating pregnancy, childbirth and the postpartum period ([Table 2](link)).

## Discussion

The data mining technique employed in this study identified causes that were not only independently linked to mortality but also that, in combination with other causes, increased the risk of maternal death. Therefore, the technique highlighted preventable causes that, although historically not primary contributors to maternal death, worsened the woman's prognosis when coexisting with other morbidities.

In this study, we found that non-obstetric causes, hypertensive disorders, pregnancy-related infections, and obstetric haemorrhage were the diagnoses most frequently associated with death during hospitalization, whether isolated or combined. We also found that abortion without complications and gestational diabetes were not associated with maternal death alone, but had an increased risk of death in the presence of other diagnoses, such as other obstetric conditions.

Direct obstetric diagnoses are the most common cause of death in the country [1]. However, the group of non-obstetric causes was the most frequent group of morbidities, with very high lifts, especially for women hospitalized for abortion, non-gestational diabetes and other obstetric conditions that are reasons for obstetric hospitalization. This result indicates the need for greater surveillance of women with non-obstetric complications, due to the greater risk of death from these conditions themselves, and also due to the possibility of worsening conditions that in isolation would have a low risk of mortality.

The interpretation of these results and their comparison with the causes of maternal death must consider the particularities of the different information systems. In the Mortality Information System, maternal deaths are classified as those occurring during pregnancy or up to the 42nd day after the end of pregnancy, due to any cause related to or aggravated by pregnancy or measures in relation to it, but not due to accidental or incidental causes. Furthermore, the definition of the cause of death occurs after the investigation of maternal death, using the cause that originated the sequence of events that resulted in death [2]. In the SIH/SUS, we analyzed all hospitalizations where a complication was recorded, not only deaths, regardless of whether this complication was the main diagnosis during hospitalization or an associated cause Furthermore, information on the date of the end of pregnancy was not available, and it is possible that deaths occurring after the 42nd day were included. Finally, external causes were also included in the morbidity analysis. Although these differences make it difficult to compare the SIM and SIH/SUS databases, they provide additional information on events that occurred after the 42nd day of the end of pregnancy and on external causes that may contribute to monitoring maternal morbidity and mortality.

Despite these differences, we observed that the main causes of death in the country and the state of São Paulo – hypertensive causes, obstetric haemorrhage and infections – were among the most frequently recorded diagnosis groups, with lifts above three, which shows that death was three times more common in women who presented these complications. These results are consistent with the national maternal mortality profile [1] and reinforce the need for improved surveillance of these complications.

The comparison of our results with those of other maternal morbidity studies is limited by how morbidity is measured, an aspect already highlighted by other authors [13]. In this study, only diagnoses, rather than medical procedures, were included. In addition, the analysis approach uses data mining techniques to identify the causes that alone or together increase the risk of maternal death. We did not identify studies in the literature that have adopted this same analysis strategy.

Brazilian [14–16] and international [17] morbidity studies that adopted the World Health Organization [18] criteria for identifying cases of maternal near miss used clinical, laboratory and management markers of organic dysfunction without identifying the underlying cause. The criteria used by the WHO for "potentially life-threatening conditions", also called severe maternal morbidity, are based on management severity indicators and diagnoses of haemorrhagic, hypertensive and other systemic disorders [18]. Haemorrhages and hypertension are related to direct obstetric causes. Moreover, other systemic diagnoses include conditions such as sepsis, shock, pulmonary embolism, convulsions and respiratory failure, which do not allow the determination of whether the origin of the disorder was an obstetric or non-obstetric cause. Therefore, the contribution of non-obstetric causes becomes less evident.

The use of morbidity scores based on the existence of preexisting conditions or those diagnosed during pregnancy has been proposed to predict severe maternal morbidity [19,20]. Different conditions receive different weights depending on their association with the outcome, and higher scores show a more significant association with the outcome. One of the studies reported an increase in the proportion of pregnant women with a score greater than or equal to 1 in the USA. One of the explanations is the greater prevalence of chronic conditions in pregnant women, the presence of which, alone or in combination with other morbidities, affects pregnancy and childbirth management [19].

SIH/SUS has limitations for evaluating maternal mortality. We have excluded 222 in-hospital deaths registered in the SIH/SUS from our analysis because these deaths were not identified in the highly reliable SIM database [21]. Incorrect death registrations in SIH-SUS might account for these discrepancies, potentially stemming from misclassification, such as recording a foetal or neonatal death in place of a maternal death. The cause of death was present in only 13.5% of the cases and was not subject to investigation. Approximately 23.7% of deaths occurred in women whose complications were not recorded. In other words, women who were hospitalized for obstetric care with no record of morbidity but who died. This result suggests the occurrence of acute complications during hospital admission without their registration in the system. We also found very high lifts in women in group 9 (unknown causes), probably recorded in deaths with a short period of hospital stay, preventing a more precise diagnostic assessment. Data from a systematic review including 26 studies and 8,704,230 women reinforce this hypothesis by showing that 48.9% of postpartum maternal deaths occurred in the first 24 hours after birth, mainly in low- and middle-income countries [22]. Most deaths from postpartum haemorrhage and pulmonary embolism occur in the first 24 hours postpartum (79.0% and 58.2%, respectively). On the other hand, deaths from hypertensive complications occur mainly in the first week postpartum, and deaths from infections occur between the 8th and 42nd day postpartum [22].

As positive aspects of the study, we highlight the use of the SIH/SUS, the only Brazilian information system with obstetric morbidity data, but which has been rarely used for the surveillance of maternal morbidity, a recommended strategy for improving obstetric care. We includedalmost 3 million publicly funded obstetric hospitalizations that occurred in the largest state of the country; we adopted a hospital episode approach, grouping multiple events of the same patient; we conducted a comprehensive analysis of all recorded diagnoses; and we useda machine learning technique to identify associations.

## Conclusion

The results of this study reaffirm the importance of hypertensive, haemorrhagic and infectious causes for maternal death and highlight the relevance of non-obstetric causes for the surveillance of maternal morbidity. This was possible by the use of the machine learning technique, which helped to identify conditions that contribute to maternal death and are likely

underestimated. It is essential to identify women with these complications, which, alone or associated with other complications, are at greater risk of death.

For SIH/SUS to be used to monitor maternal morbidity, continuous improvement of the data registry is necessary, with adequate completion of all ICDs for complications occurring during hospitalization, as well as deaths and their causes. Future studies should evaluate the importance of the main causes of hospitalization and investigate the sequence of complications that result in death, contributing to a better understanding of cases and the definition of alert systems.

## Supporting information

**S1 Table. Diagnostic groups of hospitalizations adapted using the WHO classification causes of maternal death as reference.**
(DOCX)

**S1 Fig. Heatmap of rule lifts related to death, with 2 diagnostic groups. Brazil, São Paulo state, 2014–2019.**
(TIFF)

**S2 Fig. Heatmap of rule counts related to death, with 2 diagnostic groups. Brazil, São Paulo state, 2014–2019.**
(TIFF)

## Author contributions

**Conceptualization:** Rejane Sobrino Pinheiro, Marcos Augusto Bastos Dias, Valeria Saraceni, Claudia Medina Coeli, Rosa Maria Soares Madeira Domingues.

**Data curation:** Rejane Sobrino Pinheiro, Luís Guilherme Buteri Alves, Luís Carlos Torres Guillén, Lana dos Santos Meijinhos.

**Formal analysis:** Rejane Sobrino Pinheiro.

**Funding acquisition:** Rosa Maria Soares Madeira Domingues.

**Investigation:** Rejane Sobrino Pinheiro, Marcos Augusto Bastos Dias, Natália Santana Paiva, Valeria Saraceni, Claudia Medina Coeli, Rosa Maria Soares Madeira Domingues.

**Methodology:** Rejane Sobrino Pinheiro, Marcos Augusto Bastos Dias, Natália Santana Paiva, Valeria Saraceni, Claudia Medina Coeli, Rosa Maria Soares Madeira Domingues.

**Project administration:** Rosa Maria Soares Madeira Domingues.

**Visualization:** Natália Santana Paiva.

**Writing – original draft:** Rejane Sobrino Pinheiro, Marcos Augusto Bastos Dias, Valeria Saraceni, Claudia Medina Coeli, Rosa Maria Soares Madeira Domingues.

**Writing – review & editing:** Rejane Sobrino Pinheiro, Marcos Augusto Bastos Dias, Luís Guilherme Buteri Alves, Luís Carlos Torres Guillén, Lana dos Santos Meijinhos, Natália Santana Paiva, Valeria Saraceni, Claudia Medina Coeli, Rosa Maria Soares Madeira Domingues.

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
