## [Decision Letter · Decision Letter 0]

8 Jun 2025

Dear Dr. Pinheiro,

Thank you for submitting your manuscript to PLOS ONE. After careful consideration, we feel that it has merit but does not fully meet PLOS ONE’s publication criteria as it currently stands. Therefore, we invite you to submit a revised version of the manuscript that addresses the points raised during the review process.

Thank you for your valuable contribution to the field. The reviews have provided pertinent insights that can enhance the overall quality of your manuscript and increase its novelty within the discipline.

Please pay particular attention to the following highlighted recommendations, which are essential for strengthening your work:

The manuscript lacks a clear justification for the exclusion of data from the years 2020 to 2023. An explicit explanation of this decision is necessary to ensure transparency and rigor.It is imperative to emphasize, both in the discussion and the conclusion, the use of data mining techniques as the primary methodological contribution. The authors should elaborate in greater detail why this method advances beyond existing alternatives, thereby substantiating that its application is both objective and integral to the study’s problem.While the overall study appears methodologically sound, the clarity of the language needs improvement. Enhancing the writing quality will facilitate better comprehension and engagement with your work.Regarding Table 1, please specify the study population explicitly in the title. Additionally, consider clarifying what the other categories entail. Furthermore, note that the supplementary tables are somewhat difficult to interpret visually.

I trust these suggestions will be helpful as you refine your manuscript. I look forward to seeing the improved version of your work.

We look forward to receiving your revised manuscript.

Kind regards,

Luísa da Matta Machado Fernandes, DrPH

Academic Editor

PLOS ONE

 [This work was supported by the Bill & Melinda Gates Foundation [INV-027961] and the Brazilian Ministry of Health/DECIT/CNPq [445116/2020-0]. Under the grant conditions of the Foundation, a Creative Commons Attribution 4.0 Generic License has already been assigned to the Author Accepted Manuscript version that might arise from this submission

RMSMD received the grants. RSP [3167552021-4] and CMC [306668/2023-8] are supported by the Conselho Nacional de Desenvolvimento Científico e Tecnológico

CNPq.]. 

5. We notice that your supplementary figures are uploaded with the file type 'Figure'. Please amend the file type to 'Supporting Information'. Please ensure that each Supporting Information file has a legend listed in the manuscript after the references list.

Reviewers' comments:

Reviewer's Responses to Questions

**Comments to the Author**

1. Is the manuscript technically sound, and do the data support the conclusions?

Reviewer #1: Yes

Reviewer #2: Yes

2. Has the statistical analysis been performed appropriately and rigorously?

Reviewer #1: I Don't Know

Reviewer #2: Yes

3. Have the authors made all data underlying the findings in their manuscript fully available?

Reviewer #1: Yes

Reviewer #2: Yes

4. Is the manuscript presented in an intelligible fashion and written in standard English?

Reviewer #1: No

Reviewer #2: Yes

Reviewer #1: See document attached with comments

See document attached with comments

See document attached with comments

See document attached with comments

See document attached with comments

See document attached with comments

Reviewer #2: O texto está muito bem escrito, e as análises foram conduzidas de maneira adequada. No entanto, destaco dois pontos relevantes que comprometem o alcance da pesquisa:

1 – Série histórica

Diversos estudos abordam as causas relacionadas à mortalidade materna no Brasil, frequentemente utilizando séries históricas mais extensas. A utilização de apenas seis anos de dados representa uma limitação significativa, especialmente considerando que há dados disponíveis e atualizados até 2023 — e, inclusive, dados de 2024 que, embora ainda sujeitos a revisões, já podem ser incluídos com as devidas ressalvas. O texto carece de uma justificativa clara para a exclusão dos dados referentes aos anos de 2020 a 2023.

2 – Técnica de Análise

Embora o diferencial do estudo seja a aplicação de uma técnica analítica inovadora, falta conferir maior destaque à relevância dessa escolha metodológica. A conclusão do estudo aborda as limitações das fontes de dados e apresenta os principais achados, conforme proposto no protocolo de pesquisa; no entanto, é necessário reforçar o foco no objetivo central do trabalho, que não é apenas identificar o perfil de adoecimento, mas sobretudo apresentar uma abordagem inovadora de análise.

A identificação do perfil das causas de óbitos maternos, por si só, não constitui uma inovação — inúmeros estudos já realizam essa análise, seja em recortes geográficos específicos (municípios, estados, regiões), seja em âmbito nacional. Tais pesquisas, em geral, também relacionam a mortalidade materna a fatores como o acesso aos serviços de saúde, as condições socioeconômicas e as desigualdades regionais. Para que este estudo realmente se configure como inovador, é imprescindível dar ênfase, na discussão e na conclusão, ao uso da técnica de mineração de dados como principal contribuição metodológica.

**Do you want your identity to be public for this peer review?** For information about this choice, including consent withdrawal, please see our Privacy Policy

Reviewer #1: No

Reviewer #2: No

---

## [Author Response · Author response to Decision Letter 1]

11 Jul 2025

Dear Editor,

We thank the reviewers for their careful reading and insightful comments on our manuscript, PONE-D-25-13894, “Maternal morbidity profile in hospitalizations in the Unified Health System in São Paulo, Brazil: analysis using data mining, 2014 to 2019”.

Our point-by-point response to their comments is detailed below. All modifications are highlighted in the revised manuscript.

We hope to have met all recommendations and we are available for further clarification, if necessary.

Reviewer #1:

This manuscript advances the field of women's health by proposing to analyze the profile of morbidities associated with maternal death registered in the SIH/SUS, a public in hospital information system in Brazil. The study of maternal morbidity is an essential component of maternal health surveillance, contributing to improving obstetric care, as it is more frequent than maternal deaths and shares the same determinants, as stated by the authors. In addition, the manuscript uses robust, population-based data, which is often underutilized, combined to the data mining method applied to identify diagnosis groups associated with hospital discharge (alive or death).

Overall, the manuscript is interesting, well-written, presents the main results clearly and precisely, and is somewhat innovative. Below are some points for improvement to make it more suitable for publication in Plos One.

General considerations:

• While the study appears to be sound, the language is unclear, making it difficult to follow. I advise the authors improve the writing quality. As an example, some parts are very literal and confusing, compromising readability.

Response: We would like to thank the reviewer for the careful review of our manuscript. We have reviewed all the text, including the parts indicated by the reviewer, to improve clarity. All modifications are highlighted in the manuscript.

• The excerpt from the introduction “L.66 In Brazil, approximately 80% of births are publicly funded [3]...” explains well what is not quite clear in the summary, as it seems that only 80% of births are recorded. Review the writing in the summary.

Response: The abstract was modified.

In Brazil, approximately 80% of births are publicly funded and are registered in the Hospital Information System of the Unified Health System (SIH/SUS).

• Also take a look at the Methods section L.79 “...before the start of the COVID-19 pandemic”.

Response: we have included more information about the definition of the study period.

The definition of the study period was due to the COVID-19 pandemic, which began in March 2020. The COVID-19 pandemic changed the maternal mortality profile in the country, with an increase in the maternal mortality ratio and in the proportion of deaths from non-obstetric causes, therefore being an atypical period [5,6]. We opted, therefore, to gather data from the more recent years before the COVID-19 pandemic, characterized by a largely stable maternal death pattern, to construct a dataset with an adequate number of deaths for robust data analysis.

• Or in the Discussion L.218-219, p.14 “owing to the greater risk of worsening conditions that in themselves do not pose a high risk of death and owing to the increased risk in women with other morbidities”

Response: the sentence was rewritten to make it clearer

This result indicates the need for greater surveillance of women with non-obstetric complications, due to the greater risk of death from these conditions themselves, and also due to the possibility of worsening conditions that in isolation would have a low risk of mortality.

Moreover, some excerpts present previous evidence, including data, without adequately presenting the source/reference. Here is some of what was identified:

1) The first two sentences of the introduction (L.49-51) contain information/evidence and no references.

Response: we have included the reference (reference 1) at the end of the first two sentences of the introduction

2) The same applies to lines 61-62.

Do not use a reference at the end of the paragraph, but at each previous piece of evidence presented. Revise.

Response: we have included a new reference which indicates that the investigation of maternal deaths in Brazil started in 2009.

3) Concepts on lines 124-133 requires a reference.

Response: We added a reference for the lift equation and the is.redundant function website.

• In the discussion, lines 221-226, p. 14 requires referencing it.

Response: We added the reference [2].

• In line 214 too. And so on…

Response: We added the reference [1] to this sentence.

• Specific considerations:

Introduction: In the first paragraph, the authors highlight the high rate of maternal mortality. However, in the 3rd paragraph they state that “maternal deaths are infrequent”. It would be interesting to revise the writing so that these ideas are clearer and do not suggest contradiction. The authors should clarify the following sections to avoid confusion.

Response: We rewrote the sentence, highlighting that the MMR is high, but that maternal death is an infrequent event. In fact, this is one of the difficulties in studying maternal mortality and the main justification for studying maternal morbidity. The revised text is:

The investigation of maternal deaths was established in Brazil in 2009 and is essential for a better understanding of the causes and determinants of maternal deaths [2]. However, although the MMR is high in the country, maternal deaths are infrequent events, especially in the context of health services, and it is recommended that maternal morbidity be studied in a complementary manner to develop strategies to improve obstetric care and reduce maternal mortality [2].

• The title, objective in the abstract and objective in the introduction of the study highlight the method (data mining). Sometimes this is more prominent than the objective of the study itself. Is this really necessary? If so, the authors will need to present in more detail why this method advances on other existing methods, in order to argue that the use of this method is also objective and part of the study problem. Also, the objective of the abstract and the introduction are very different, I suggest standardizing them.

Response: We appreciated the reviewer's insightful observations and helpful recommendations.

The rewritten paragraph follows:

Studies on maternal health during pregnancy, delivery, and postpartum have traditionally examined the isolated effects of various causes on maternal mortality. However, morbidities may operate independently or exhibit interconnections that impact women's health differently. Interpreting statistical models becomes even more challenging when analyzing rare events when numerous morbidities might interact or occur in isolation, leading to a high number of independent variables. By employing data mining techniques, analysis strategies can yield novel insights from extensive variable sets, especially those contributing to an increased risk of maternal death. Understanding how distinct morbidity profiles relate to higher risks of maternal death is crucial for supporting prevention efforts and identifying early warnings of adverse prognosis.

We standardize the objective in the abstract and at the end of the introduction:

Thus, the objective of this study was to apply data mining techniques to identify diagnoses registered in the SIH/SUS that alone or together are associated with maternal death

• Methods: Authors declare they have “excluded 222 hospital episodes with “death in hospital” as the reason for discharge because we could not identify deaths in the mortality database via a deterministic matching algorithm.”, but do not recall this limitation in the discussion section stating how could this affect their results. Couldn't this exclusion of 222 episodes bring an important bias to the research, since maternal death is a rare event?

Response: The exclusion of these deaths was based on the low probability of their occurrence, given their absence from the highly reliable death database (SIM). Incorret death registrations in SIH-SUS might account for these discrepancies, potentially stemming from misclassification, such as recording a neonatal death in place of a maternal death. We have included this explanation in the discussion section. As follows:

SIH/SUS has limitations for evaluating maternal mortality. We have excluded 222 in-hospital deaths registered in the SIH/SUS from our analysis because these deaths were not identified in the highly reliable SIM database [21]. Incorrect death registrations in SIH-SUS might account for these discrepancies, potentially stemming from misclassification, such as recording a foetal or neonatal death in place of a maternal death.

• Moreover, the procedure “All AIH records of the same patient identified by the algorithm were grouped and analyzed as one hospital episode.” is not very clear, because couldn't a woman have had several hospitalizations in the period studied, with different AIHs, and from different pregnancies? How were these repetitions treated? What was their frequency? Was it 1.1%? (L101, p.5?) Clarify.

Response: As informed in the text, we only grouped AIH as part of the same episode of care if the subsequent record occurred within a one-day maximum difference between the admission date and the previous discharge date. However, to increase the clarity of the text, we included the text in bold:

“When AIH discharge was a hospital stay or an interhospital transfer, the subsequent record (within a one-day maximum difference between the admission date and the previous discharge date) was searched via a deterministic record linkage algorithm using variables available in deidentified databases: the hospital code, the municipality of residence, and the patient's date of birth. All AIH records of the same patient identified by the algorithm as belonging to the same hospitalization were grouped and analysed as one hospital episode of care.

Additionally, we added a reference that explains the development of this algorithm: Coeli CM, Domingues, Meijinhos L, Bastos DMC, Pinheiro RS, Saraceni V, Dias MAB, Paiva NS, Camargo Jr KR. Using a deterministic matching computer routine to identify hospital episodes in a Brazilian de-identified administrative database for the analysis of obstetrics hospitalisations. International Journal of Population Data Science (2024) 10:1:09.

• “…the WHO maternal ICD groups were adapted to capture all diagnoses recorded during…” � how was the adaptation?

Response: The WHO maternal ICD groups are used for the classification of maternal deaths. As we are studying maternal morbidity, we added ICD codes that are not considered causes of death but are causes of morbidity and hospital admissions to the existing groups and also created new groups that are exclusively related to maternal morbidity. The adaptation was proposed by an obstetrician and is displayed in S1 Table. We included a new text in the methods section giving more details about this adaptation.

“In this adaptation, proposed by an obstetrician of our research team, ICD codes that are not considered causes of death, but are causes of morbidity and hospital admissions, were added to the existing maternal ICD groups, while new groups that are exclusively related to maternal morbidity were created. The proposed adaptation is displayed in S1 Table.”

• Results: In table 1, identify the study population in the title What would be others?

Response: We corrected the title of Table 1.

Table 1. Frequency of diagnostic groups in obstetric hospitalizations of women aged 10-49 years with death in SIH/SUS. State of São Paulo, Brazil, 2014-2019.

We added a footnote to Table 1 explaining the "Others" category:

*Note: "Other" group includes all ICD codes that do not fall into the previously specified categories.

• The supplementary tables are difficult to see.

Responde: We restructured and revised the wording of certain diagnostic groups/subgroups within Table 1. Additionally, we ensured consistency between the manuscript text and the content of the other tables.

• Discussion: In the third paragraph, authors established differences between their results with the SIM results. But it seems like they lack pointing out how the additional information after 42nd day and external causes that were included might advance in monitoring maternal morbidity and mortality….

Response: We would like to thank the reviewer for this insightful comment. We have added the possible improvements of including deaths after 42nd day and external causes in the text.

“Although these differences make it difficult to compare the SIM and SIH/SUS databases, they provide additional information on events that occurred after the 42nd day of the end of pregnancy and on external causes that may contribute to monitoring maternal morbidity and mortality.”

• Authors point the result of the study as follows “We also found that abortion and diabetes were associated with maternal death in the presence of other diagnoses, such as nonobstetric causes”, but do not discuss it, specifically. This could be quite relevant because it is potentially prevented.

Response: In the first paragraph of the discussion section, we only presented the most relevant results of our study. The topic mentioned above is discussed in the second paragraph of the discussion section, where we reinforce the importance of the non-obstetric causes of morbidity in isolation or in combination with other causes, such as abortion and diabetes, increasing the risk of maternal death.

• Authors should also consider to write the implications of the results found, which aligns with the ones of Maternal Mortality causes of death, but also advances showing other results, including combinations of diseases and causes, along with the great amount of nonidentified causes…

We added a new text to the manuscript's discussion and conclusion emphasizing disease combinations, thus reinforcing the findings highlighted in the results section.

In the discussion section:

The data mining technique employed in this study identified causes that were not only independently linked to mortality but also that, in combination with other causes, increased the risk of maternal death. Therefore, the technique highlighted preventable causes that, although historically not primary contributors to maternal death, worsened the woman's prognosis when coexisting with other morbidities.

In conclusion section:

The data mining technique employed in this study highlighted conditions that contribute to maternal death and are likely underestimated.

• What is the reference on the L267-270?

Response: To bring the citation closer to the assertion it supports, we moved the reference from the end of the paragraph to the end of the first sentence.

• The first paragraph of conclusion is not the conclusion itself. I recommend to include it to the discussion section when highlighting the relevance of the study.

Response. Thank you for the suggestion. We moved this paragraph to the discussion section.

Reviewer #2:

1) O texto está muito bem escrito, e as análises foram conduzidas de maneira adequada. No entanto, destaco dois pontos relevantes que comprometem o alcance da pesquisa:

1 – Série histórica

Diversos estudos abordam as causas relacionadas à mortalidade materna no Brasil, frequentemente utilizando séries históricas mais extensas. A utilização de apenas seis anos de dados representa uma limitação significativa, especialmente considerando que há dados disponíveis e atualizados até 2023 — e, inclusive, dados de 2024 que, embora ainda sujeitos a revisões, já podem ser incluídos com as devidas ressalvas. O texto carece de uma justificativa clara para a exclusão dos dados referentes aos anos de 2020 a 2023.

Response: We would like to thank the reviewer for this comment. We opted to exclude the years from 2020 to 2023 from our analysis to mitigate the influence of transient health conditions and outcomes observed during the COVID-19 pandemic and the subsequent period. We instead gathered data from more recent years, characterized by a largely stable maternal death pattern, to construct

---

## [Decision Letter · Decision Letter 1]

10 Sep 2025

Maternal morbidity profile in hospitalizations in the Unified Health System in São Paulo, Brazil: analysis using data mining, 2014 to 2019

PONE-D-25-13894R1

Dear Dr. Pinheiro,

We’re pleased to inform you that your manuscript has been judged scientifically suitable for publication and will be formally accepted for publication once it meets all outstanding technical requirements.

Kind regards,

Luísa da Matta Machado Fernandes, DrPH

Academic Editor

PLOS ONE

Additional Editor Comments (optional):

Reviewer #2:

Reviewers' comments:

Reviewer's Responses to Questions

**Comments to the Author**

Reviewer #2: All comments have been addressed

2. Is the manuscript technically sound, and do the data support the conclusions?

Reviewer #2: Yes

3. Has the statistical analysis been performed appropriately and rigorously?

Reviewer #2: Yes

4. Have the authors made all data underlying the findings in their manuscript fully available?

Reviewer #2: Yes

5. Is the manuscript presented in an intelligible fashion and written in standard English?

Reviewer #2: Yes

Reviewer #2: (No Response)

**Do you want your identity to be public for this peer review?** For information about this choice, including consent withdrawal, please see our Privacy Policy

Reviewer #2: No

---

## [Editor Report · Acceptance letter]

PONE-D-25-13894R1

PLOS ONE

Dear Dr. Pinheiro,

I'm pleased to inform you that your manuscript has been deemed suitable for publication in PLOS ONE. Congratulations! Your manuscript is now being handed over to our production team.

Kind regards,

on behalf of

Dr. Luísa da Matta Machado Fernandes

Academic Editor

PLOS ONE